# Survey on Dermatophytes Isolated from Animals in Switzerland in the Context of the Prevention of Zoonotic Dermatophytosis

**DOI:** 10.3390/jof9020253

**Published:** 2023-02-14

**Authors:** Marina Fratti, Olympia Bontems, Karine Salamin, Emmanuella Guenova, Michel Monod

**Affiliations:** 1Service de Dermatologie, Laboratoire de Mycologie, Centre Hospitalier Universitaire Vaudois (CHUV), 1011 Lausanne, Switzerland; 2Faculty of Biology and Medicine, University of Lausanne, 1015 Lausanne, Switzerland

**Keywords:** *Trichophyton mentagrophytes*, *Trichophyton benhamiae*, *Microsporum canis*, *Nannizzia gypsea*, *Nannizzia persicolor*, epidemiology, dermatophytosis, cats, dogs, guinea pigs

## Abstract

Most inflammatory dermatophytoses in humans are caused by zoophilic and geophilic dermatophytes. Knowledge of the epidemiology of these fungi in animals facilitates the prevention of dermatophytosis of animal origin in humans. We studied the prevalence of dermatophyte species in domestic animals in Switzerland and examined the effectiveness of direct mycological examination (DME) for their detection compared to mycological cultures. In total, 3515 hair and skin samples, collected between 2008 and 2022 by practicing veterinarians, were subjected to direct fluorescence microscopy and fungal culture. Overall, 611 dermatophytes were isolated, of which 547 (89.5%) were from DME-positive samples. Cats and dogs were the main reservoirs of *Trichophyton mentagrophytes* and *Microsporum canis*, whereas *Trichophyton benhamiae* was predominantly found in guinea pigs. Cultures with *M. canis* significantly (*p* < 0.001) outnumbered those with *T. mentagrophytes* in DME-negative samples (19.3% versus 6.8%), possibly because *M. canis* can be asymptomatic in cats and dogs, unlike *T. mentagrophytes*, which is always infectious. Our data confirm DME as a reliable, quick, and easy method to identify the presence of dermatophytes in animals. A positive DME in an animal hair or skin sample should alert people in contact with the animal to the risk of contracting dermatophytosis.

## 1. Introduction

Dermatophytes are the most common pathogenic agents of superficial mycoses, infecting almost exclusively the *stratum corneum*, nails, and hair [1,2]. Dermatophytoses are the most common skin diseases worldwide, and their prevalence is probably underestimated [3]. While localized lesions of the glabrous skin can be treated topically, systemic treatment is generally necessary for extensive infections, tinea capitis, and tinea unguium [4,5]. Three broad ecological groups of dermatophyte species are recognized, namely anthropophilic, zoophilic, and geophilic species, depending on their natural reservoir [2,6,7]. Anthropophilic species naturally colonize humans, whereas zoophilic species are predominantly found in animals. Geophilic species in the soil may sporadically cause disease, but in general, they are non-pathogenic saprophytes. Similar to dermatophytosis in humans, the clinical appearance of dermatophytosis in animals varies largely. Dermatophytes in animals often cause a ring-shaped rash, called ringworm. The characteristics are circular or irregular alopecic lesions with scaling or crusting, generalized alopecia, kerion, paronychia, and claw infections [2,7]. *Microsporum canis* in cats and *Trichophyton benhamiae* in guinea pigs also cause chronic infections with minor lesions, discernible only on close examination, and many animal carriers are asymptomatic [8,9,10]. Most inflammatory dermatophytoses of the skin, beard, and hair in humans are caused by zoophilic and geophilic species [11]. An overshooting immune response to ringworm infection, called kerion, is a common clinical phenomenon in tinea capitis and tinea barbae [12]. Although variable, *M. canis* infections are less inflammatory than those caused by other zoophilic species and rarely cause severe inflammation [2,13]. Dermatophytosis is considered one of the most common zoonotic diseases.

In cases of inflammatory dermatophytosis in patients, mainly tinea corporis, tinea faciae, and tinea capitis, it is important to identify the possible source of animal transmission to prevent recurrence, family outbreaks, or rapidly progressing epidemics. Here, we present a survey on dermatophytes in Switzerland, including all species isolated from animal lesions referred to the mycology laboratory of the Department of Dermatology of the University Hospital of Lausanne (CHUV) from 2008 to 2022. It reveals the current trends in the epidemiology of dermatophyte infections in Switzerland and the common reservoirs of zoophilic fungal species as a possible source of infection. One of the objectives of this study was also to evaluate direct mycological examination (DME) as a rapid and reliable method for the detection of dermatophytes in hair and scales of pets.

## 2. Material and Methods

### 2.1. Animal Samples

Dermatological samples were collected by practicing veterinarians from symptomatic and asymptomatic animals by taking hair and skin scales or using toothbrushes. The collected material was enclosed in a labeled plastic bag or a small plastic box and sent to the mycology laboratory of the Department of Dermatology at the CHUV for mycological analysis.

### 2.2. Sample Processing

Each sample collected was subjected to direct fluorescence microscopy and fungal culture in Sabouraud agar medium with actidione. A first sample portion was examined in a dissolving solution with fluorochrome. The solution was prepared by dissolving 1 g of sodium sulfide (Na_2_S) (Sigma, St. Louis, MO, USA) in 7.5 mL of distilled water and subsequently adding 2.5 mL of ethanol. Thereafter, 10 μL of blankophor (Indulor Leverkusen, Ankum, Germany) was added to this mixture [14,15]. Animal hairs and scales were placed on a slide with two to three drops of dissolving and staining solution and covered with a coverslip. After at least 1 min of incubation, the slides were examined using a Zeiss Axioskop fluorescence microscope with excitation between 360 and 400 nm or between 400 and 440 nm.

A second portion of each sample was used to perform culture assays in test tubes containing slanted Sabouraud agar medium with chloramphenicol (50 μg/mL) and actidione (400 μg/mL) (BBL Mycosel agar, Becton Dickinson, Sparks, MD, USA). Either hair fragments and scales were deposited at the surface of the gelose or the toothbrush was gently pushed into the agar, followed by brushing it thoroughly over the agar medium. The inoculated tubes were incubated at 30 °C, and dermatophytes were identified after 10–14 days of growth by macroscopic and microscopic examination. The nomenclature of dermatophytes in de Hoog et al., 2019, was adopted [16]. We used *T. mentagrophytes* for the species of the *T. mentagrophytes/interdigitale* complex. Many strains of this complex are indistinguishable on the basis of phenotypic characters, but *Trichophyton interdigitale* is almost always isolated from tinea pedis or tinea unguium and is a distinct species [17,18]. We did not distinguish *Nannizzia gypsea* (formerly *Microsporum gypseum*), *Nannizzia fulva*, and *Nannizzia incurvata*, which are three closely related geophilic species producing numerous spindle-shaped macroconidia [19,20]. Instead, we agreed on using *Nannizzia gypsea* for the analysis reports.

When the identification of dermatophyte species based on their morphological appearance in culture and microscopic observations was doubtful, identification was performed using molecular biology. The 28S rRNA gene and the internal transcribed spacer (ITS) were amplified using the oligonucleotide pairs 5′-GATAGCGMACAAGTAGAGTG-3′/5′-GTCCGTGTTTCAAGACGGG-3′ (LSU1/LSU2) and 5′-GGTTGGTTTCTTTTCCT-3′/5′-AAGTAAAAGTCGTAACAAGG-3′ (ITS1/ITS2 or LR1/SR6R), respectively [17,18,21,22]. Species identification was performed using BLAST analysis of PCR product sequences, similar to the mycological analysis of human dermatophytosis [23].

## 3. Results

From 2008 to 2022, the total number of samples sent for mycological analysis was 3515 (Table 1). Most samples were from pets, namely cats, dogs, and guinea pigs; 15 samples were from other mammals, including cattle and horses, and 12 samples were from reptiles. The DME often showed sleeves of septate filaments or round arthrospores in strings of pearls around the hairs (Figure 1A,B), multiple isolated septate filaments, or multiple dispersed round spores. In these cases, DME was considered as positive for fungal infection. In toothbrush samples from asymptomatic animals, we also frequently detected pellet-like accumulations of twisted hyphae, which we designated as “nodes” (Figure 1C,D), and DME was deemed positive. The mere presence of isolated spores was deemed a negative DME result as animals are often in contact with soil, grass, or straw.

Dermatophytes were isolated from 611 collected samples (Table 1). Positive DME was revealed in 547 (89.5%) of these samples. Only 64 dermatophytes (10.5%) were obtained from samples when DME was negative. Of a total of 3515 samples analyzed, 223 samples with positive DME (6.3%) did not generate a dermatophyte mycelium in cultures, and 2681 samples (76.3%) were negative for both DME and dermatophyte growth.

### 3.1. Isolated Dermatophyte Species

*Trichophyton mentagrophytes* was the most frequently isolated species (Table 2). Its prevalence in relation to the number of dermatophytes isolated from cats and dogs was 63.8% (N = 233/365) and 63.0% (N = 114/181), respectively. This species was also isolated from one guinea pig, two rabbits, and a horse. It encompasses various genotypes which were not identified in our routine analyses. PCR identification was performed on a total of 51 strains when identification based on phenotypic characters was doubtful or could be confused with a *T. benhamiae* of white phenotype. The dominant genotype of strains from cats and dogs we studied further was type III [24], with a 28S sequence identical to AF378740 and an ITS sequence identical to AF506034 [17]. One strain from a dog had an ITS sequence identical to GU646879 and was deposited in the IHEM collection as IHEM22711 [20]. The two strains isolated from rabbits had a 28S sequence identical to AF378740 and an ITS sequence identical to GU646874 [25]. The species *M. canis* grew less often isolated than *T. mentagrophytes* from cats and dogs. Its prevalence in relation to the number of dermatophytes isolated from each of these two animal species was 35.6% (N = 130/365) and 22.1% (N = 40/181), respectively.

Among the 50 dermatophytes isolated from guinea pigs, 48 were *T. benhamiae*, with a prevalence of the yellow phenotype (N = 43) in relation to the white phenotype (N = 5). *Trichophyton benhamiae* was also isolated four times from dogs, once from a cat, once from a rabbit, and once from a rodent (degu). One strain of *T. erinacei*, which is closely related to *T. benhamiae* [16], was isolated from a hedgehog. This fungus caused a highly inflammatory ringworm on the hand of its owner [26].

Both *T. verrucosum* and *T. equinum* were identified in low numbers due to the small number of cattle and horse isolates sent to our laboratory. Samples from livestock were generally not sent to our laboratory, which explains the low number of isolated *T. verrucosum*. This species was isolated twice from samples taken from pigs at a farm.

Species of the genus *Nannizzia* (*N. persicolor* and *N. gypsea*) were almost exclusively isolated from dogs. No dermatophytes of the genera *Trichophyton, Microsporum,* or *Nannizzia* were isolated from reptiles (12 samples). However, *Nannizziopsis guarroi* (formerly *Chrysosporium guarroi*) with a 28S sequence identical to MH874904 was isolated once from a pogona (a reptile of the infraorder *Iguania*) (Figure 2). This species has been previously isolated from iguanas, pogonas, and other reptiles [27,28]. *Nannizziopsis* species belong to the *Nannizziopsiaceae* family related to that of the dermatophytes (*Arthrodermataceae*) in the order *Onygenales* [28,29].
jof-09-00253-t002_Table 2Table 2Dermatophytes isolated in Lausanne from animal samples collected by veterinarians between 2008 and 2022.
CatsDogsGuinea PigsRabbitsHorseMiscellaneousTotal***T. mentagrophytes***233114221 ^1^
**352*****M. canis***13040


1 (cheetah)**171*****T. benhamiae***14481
1 (degu)**55*****N. persicolor***16



**7*****N. gypsea***
17

21**20*****T. verrucosum***




4 (cattle); 2 (swine) 2**4*****T. equinum***



1
**1*****T. erinacei***




1 (hedgehog) ^2^**1****Total****365****181****50****3****4****8****611**^1, 2^ Published as a case report [26,30].


### 3.2. Efficiency of DME for the Detection of Dermatophytes in Animal Samples

We examined whether the prevalence of isolated dermatophytes varied by host and dermatophyte species when direct examination was negative (Table 3). A dermatophyte was isolated in culture in 87.9% (N = 321/365), 92.3% (N = 167/181), and 90.0% (N = 45/50) of cases when DME was positive in samples from cats, dogs, and guinea pigs, respectively. Considering all samples of all animals, a dermatophyte was isolated in culture in 89.5% of cases when DME was positive (N = 547/611). Interestingly, the ratio of negative DME/positive DME was significantly higher for *M. canis* [33/138 (23.9%)] than for *T. mentagrophytes* [24/328 (7.3%)], with *p* < 0.001 (Chi-square test with the data in the lower part of Table 3). In other words, *M. canis* was isolated more frequently than *T. mentagrophytes* when DME was negative.

## 4. Discussion

Three species, namely *T. mentagrophytes*, *T. benhamiae*, and *M. canis*, were the most frequently isolated dermatophytes in animals in Switzerland in the last 15 years (2008–2022). No anthropophilic species was recorded [31]. *Trichophyton mentagrophytes* was isolated mainly from cats and dogs but also from other animals. The cats with *T. mentagrophytes* are usually hunters and have skin lesions, whereas cats carrying *M. canis* are generally indoor domestic cats [32]. Therefore, *T. mentagrophytes* infections probably occur during hunting, and the source of this dermatophyte may be soil or rodents. Contrary to a recent report from the south of England [33], the prevalence of *T. mentagrophytes* on cats and dogs was higher than that of *M. canis*.

Most *T. benhamiae* isolates were from guinea pigs. The zoonotic potential of this dermatophyte in teenagers and young adults is well known and reported in studies from Switzerland, France, and Germany, where *T. benhamiae* is increasingly recognized as an emerging species source of human dermatophytosis [9,10,22,34,35]. Many guinea pig carriers are healthy and asymptomatic [9,10]. The emergence of *T. benhamiae* is due to a change in the pet owners. In many families, guinea pigs have been preferred to cats or dogs because they are less demanding and less expensive. *Trichophyton benhamiae* is closely related to *T. erinacei*, which causes highly inflammatory tinea manum (hand infections) in patients. The latter species was isolated only once, but several cases of this emerging dermatophyte species were recently reported in Spain and Germany [36,37,38].

In our cohort, we found only dogs, with one exception (one cat), that were infected with *N. persicolor*, in accordance with previous reports [39,40]. To our knowledge, this is the first time that *N. persicolor* has been isolated from a cat. In a recent survey of dermatophytes of dogs and cats in England, *N. persicolor* was not isolated from cats [33]. This dermatophyte species is considered zoophilic because it was isolated from bank voles and mice [41,42]. Interestingly, a Canadian survey by Muller et al. mentions that 12 of 16 dogs infected with *N. persicolor* were hunters [40], as for cats and *T. mentagrophytes* [31]. Human ringworms with *N. persicolor*, which are highly inflammatory [30,43,44], may originate from contact with dogs.

*Nannizzia gypsea*, which is geophilic, was also mostly isolated from dogs (17/20 isolates, Table 2). We did not identify this species in cat samples, contrary to a previous report [45]. *Nannizzia gypsea* dermatophytosis in humans originates from the soil but can also be acquired through contact with dogs [46]. Isolates of *N. gypsea* and *N. persicolor*, with two exceptions, were from DME-positive samples in our survey (Table 3).

Of all samples tested, 71% of positive DME cases (N = 770) resulted in a dermatophyte culture (N = 547) (Table 1). Only 11% of dermatophytes (N = 64/611) were obtained from DME-negative samples. These results validate the sensitivity of fluorescence microscopy for the identification of fungal elements in dermatological samples. Hyphae and spores are highly fluorescent and easily detectable (Figure 1) [11,47]. However, DME is more difficult to interpret in animal samples than in human samples because animals are in contact with soil, grass, or straw and have a much higher hair density. Therefore, the observation of a few isolated spores was considered a negative DME. Another problem is the presence of fast-growing contaminating molds in the culture tests, which can compromise the isolation of a slower-growing dermatophyte. No dermatophyte was isolated in 223 samples that were DME positive (Table 1). *M. canis* was isolated more frequently than *T. mentagrophytes* when DME was negative (19.3% versus 6.8%) (Table 3). This difference may be related to the fact that *M. canis* can be asymptomatic in cats and dogs, unlike *T. mentagrophytes*, which is always infectious.

## 5. Conclusions

The present survey of dermatophytes isolated from animal samples showed that cats and dogs are the main reservoirs for *T. mentagrophytes* and *M. canis,* and guinea pigs for *T. benhamiae* (Table 3). This indicates the importance of *T. mentagrophytes* and the considerable frequency of *M. canis* with the large number of cats and dogs as pets. Dogs also appeared to be a reservoir of *N. gypsea* and *N. persicolor*. The best preventive measure to avoid dermatophytosis in humans is to avoid direct contact with contaminated pets. Clinically, it is essential to accurately identify the fungus causative for inflammatory dermatophytosis in humans and to carefully examine pets as a possible source of infection. Fluorescence microscopy is an inexpensive, sensitive, and rapid method for detecting dermatophytes in animals. Although DME does not identify the infecting fungus, a positive DME may provide an idea of the potential dermatophyte species to incriminate, depending on the animal, and may already draw attention to the risk of contracting dermatophytosis. Fungal cultures remain indispensable for determining the dermatophyte species and may uncover a dermatophyte carrier in some cases of a negative DME. Infections in humans can be prevented by treating animals diagnosed as infected or asymptomatic carriers.

## Figures and Tables

**Figure 1 jof-09-00253-f001:**
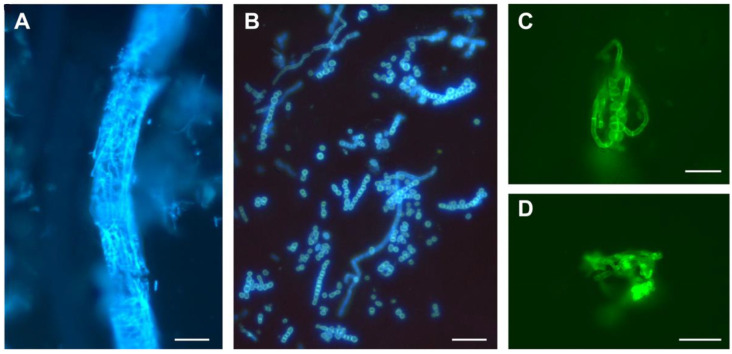
Direct mycological examinations of animal samples. (**A**) Sleeve of septate filaments around a cat hair (bar = 40 µm). (**B**) Round arthrospores in strings of pearls around an infected dog hair (bar = 20 µm). (**C**,**D**) Pellet-like accumulations of twisted hyphae from asymptomatic guinea pigs (bar = 20 µm). The sample preparations were observed with a fluorescence microscope using excitation between 360 and 400 nm (**A**,**B**), and between 400 and 440 nm (**C**,**D**).

**Figure 2 jof-09-00253-f002:**
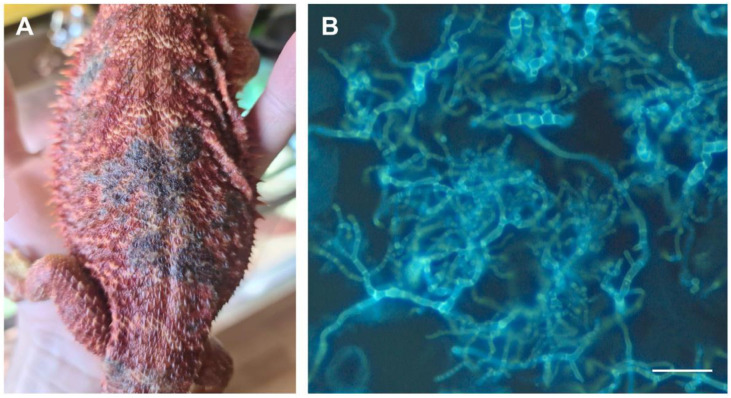
*Nannizziopsis guarroi* in a pogona (**A**) and direct mycological examination (**B**). The sample preparations were observed with a fluorescence microscope using excitation between 360 and 400 nm (bar = 20 µm).

**Table 1 jof-09-00253-t001:** Analysis of animal samples at the CHUV from 2008 to 2022 for the detection and identification of dermatophytes.

	Cultures with a Dermatophyte	Dermatophyte-Free Cultures	Totals
DME positive	547 (15.6%)	223 (6.3%)	770 (21.9%)
DME negative	64 (1.8%)	2681 (76.3%)	2745 (78.1%)
Total	611 (17.4%)	2904 (82.6%)	3515 (100.0%)

Percentages are given in relation to the total number of analyses (N = 3515).

**Table 3 jof-09-00253-t003:** Direct mycological examinations per animal for each dermatophyte species isolated in culture.

Animals	Dermatophyte Species	Number of Isolates	DME Positive	DME Negative
**Cats**	*T. mentagrophytes*	233	216 (93%)	17 (7%)
	*M. canis*	130	103 (79%)	27 (21%)
	*N. persicolor*	1	1 (100%)	0 (0%)
	*T. benhamiae*	1	1 (100%)	0 (0%)
** *Subtotals for cats* **		** *365* **	** *321 (88%)* **	** *44 (12%)* **
**Dogs**	*T. mentagrophytes*	114	108 (95%)	6 (5%)
	*M. canis*	40	34 (85%)	6 (15%)
	*T. benhamiae*	4	4 (100%)	0 (100%)
	*N. gypsea*	17	15 (88%)	2 (12%)
	*N. persicolor*	6	6 (100%)	0 (100%)
** *Subtotals for dogs* **		** *181* **	** *167 (92%)* **	** *14 (8%)* **
**Guinea pigs**	*T. mentagrophytes*	2	1	1
	*T. benhamiae*	48	44 (92%)	4 (8%)
** *Subtotals for guinea pigs* **		** *50* **	** *45 (90%)* **	** *5 (10%)* **
**Rabbit**	*T. mentagrophytes*	2	2	0
	*T. benhamiae*	1	1	0
**Horses**	*T. mentagrophytes*	1	1	0
	*T. equinum*	1	1	0
	*N. gypsea*	2	2	0
**Cows**	*T. verrucosum*	2	2	0
**Degu**	*T. benhamiae*	1	0	1
**Cheetah**	*M. canis*	1	1	0
**Pigs**	*T. verrucosum*	2	2	0
**Hedgehog**	*T. erinacei*	1	1	0
	*N. gypsea*	1	1	0
**Total**		**611**	**547**	**64**
	*T. mentagrophytes*	352	328 (93.2%)	24 (6.8%)
	*M. canis*	171	138 (80.7%)	33 (19.3%)
	*T. benhamiae*	55	50 (90.9%)	5 (9.1%)
	*N. gypsea*	20	18	2
	*N. persicolor*	7	7	0
	*T. equinum*	1	1	0
	*T. verrucosum*	4	4	
	*T. erinacei*	1	1	0
**Total**		**611**	**547**	**64**

## Data Availability

Research data can be shared with the Mycology Laboratory of the Department of Dermatology of the CHUV on request.

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
