# Peer review of "Survey on Dermatophytes Isolated from Animals in Switzerland in the Context of the Prevention of Zoonotic Dermatophytosis"

_jof, 2023, doi:10.3390/jof9020253_

Round 1

Reviewer 1 Report

Please see my comments in the attached PDF file 

Author Response

Reviewer 1

Minor revision

This is an interesting report on direct mycological examination DME as a reliable, quick, and easy method that allows to identify the presence of dermatophytes in animals. It was applied in a pool of samples of domestic animals in Switzerland. The paper provides an interesting inside of the distribution of certain fungal species in domestic animals, however it may be improved at some places:

1

Line 30 - I suggest to add some epidemiology data (reference) to underline the importance to combat such diseases

Response

We added the following on line 30

Dermatophytoses are the most common skin diseases worldwide, and their prevalence is probably underestimated [3]. While lesions of the glabrous skin can be treated topically, systemic treatment is generally necessary for extensive infections and tinea capitis and tinea unguium [4,5].    

The three following references were added:

Hayette, M.P.; Sacheli, R. Dermatophytosis, trends in epidemiology and diagnostic approach. Curr Fungal Infect Rep. 2015, 9,164–179. doi 10.1007/s12281-015-0231-4.

Hay, R.J. Tinea capitis: Current status. Mycopathologia 2017, 182, 87-93. doi: 10.1007/s11046-016-0058-8.

Asz-Sigall, D.; Tosti A.; Arenas, R. Tinea Unguium: Diagnosis and treatment in practice. Mycopathologia 2017, 182, 95_100. doi: 10.1007/s11046-016-0078-4.

2

Line 53 - has this study being published? Please add a reference.

Response

In fact, we only referred to our survey. The sentence was changed to remove the ambiguity:

The sentence on line 53 is now

Here, we present a survey on dermatophytes in Switzerland including all species isolated from animal lesions referred to the mycology laboratory of the Department of Dermatology of the University Hospital of Lausanne (CHUV) from 2010 to 2022

3

Line 157 - there is some symbol visible above of the A in Figure 2 - does it have a meaning?

Response

The symbol was removed

4

Line 177 - "was" - > "were". I would add a citation here.

Response

"was" seems to us correct. A reference was added

5

Line 186 - these countries are geographically close to each other. Was there maybe any correlation observed, for example climate or temperature dependent distribution of T. benhamiae?

Response:

The following sentences have been inserted on line 195:

The emergence of T. benhamiae is due to a change in the pet owners. In many families, guinea pigs have been preferred to cats or dogs because they are less demanding and less expensive (line 194).

6

Line 194 - I suggest to provide a short but detailed discussion concerning this interesting observation. Can one exclude an artefact / mistake?

Response

  1. persicolor is an easy-to-identify dermatophyte because of its spindle-shaped macroconidia, abundant microspores and numerous tendril-forming hyphae. Consequently, one can exclude a mistake. This comment was not iserted

7

Line 231-232 - can these studies bring the combat against these pathogens forward

Response

The sentence “Infections in humans can be prevented by treating animals diagnosed as infected or asymptomatic carriers” was added at the end of Discussion

Reviewer 2 Report

Comments

The study is well conducted and contributes to the knowledge of the epidemiology of dermatophytosis in animals to improve the prevention of human dermatophytosis, as well as the recurrence of infection, or family outbreaks of veterinary origin. A large number (n=3,515) of hair and skin veterinary samples and dermatophyte isolates (n=611) were included. The authors performed a rigorous molecular analysis to define species. The results are quite interesting, showing distinct fungal species infecting different animals. Moreover, the confirm direct examination of clinic samples is a reliable, quick, and easy method to identify the presence of dermatophytes in animals. This review has a few concerns, as follows:

This review does not agree with the author's idea” Depending on the domestic animal, a positive direct examination gives an idea of the fungal species…..”. In fact, the authors affirm elsewhere: “When the identification of dermatophyte species based on their morphological appearance in culture and microscopic observations was doubtful, identification was performed using molecular biology. ” and “Fungal cultures remain indispensable to determine the dermatophyte species…”.

There is contradictory information about the study period: in Material and Methods. Animal samples it is referred to as 2010 to 2022, but in the Abstract, Results, and Tables 1,2  the authors cite 2008 and 2022.

The manuscript and results are very clear even a great variety of animal species were studied. However, I suggest indicating which animals were considered pets (for companionship or pleasure) and which were domestic animals to produce food or utility. The degree of contact between a pet owner and his animal is supposed to be higher than between a worker and animals for utility. So, the risk of acquiring a dermatophytid infection is more significant with a pet. It would be interesting a discussion about the risk of farm workers being infected by domesticated livestock (horses, cattle, and other animals for utility).

The unit system must be corrected since “ml” is wrong. According to
International System of Units, “L” is for liters, and “l” is for length.
The entire
manuscript should be reviewed.

Line 115: If 2745 samples were DME negative (Table 1) 64 represents 2.3% and not 10.5% as informed

Lines 116-118. Inform the percentages for the given number of isolates. These are the metrics that are needed.

Line 127.  Does the data of routine analyses were published elsewhere? If it was not published or is not a Supplementary Material it may be informed as Data not shown.

Line 137. It would be of interest to inform the percentage of yellow-colored cultures found among T. benhamiae cultures.

Line 153. Delete a point before the reference.

Line 175. Insert a point after T.

Figure 1D needs to focus

Figure 1. Delete “Samples were examined in a dissolving solution containing a fluorochrome that specifically binds fungal cell wall polysaccharides [11]” since it should be in Methods.

Table 3. In Column 1, delete the information “Dermatophytes species”, since it is not related to animals. Be sure that this information is in the Results section.

Author Response

Reviewer 2

Comments

The study is well conducted and contributes to the knowledge of the epidemiology of dermatophytosis in animals to improve the prevention of human dermatophytosis, as well as the recurrence of infection, or family outbreaks of veterinary origin. A large number (n=3,515) of hair and skin veterinary samples and dermatophyte isolates (n=611) were included. The authors performed a rigorous molecular analysis to define species. The results are quite interesting, showing distinct fungal species infecting different animals. Moreover, the confirm direct examination of clinic samples is a reliable, quick, and easy method to identify the presence of dermatophytes in animals. This review has a few concerns, as follows:

1

This review does not agree with the author's idea” Depending on the domestic animal, a positive direct examination gives an idea of the fungal species…..”. In fact, the authors affirm elsewhere: “When the identification of dermatophyte species based on their morphological appearance in culture and microscopic observations was doubtful, identification was performed using molecular biology. ” and “Fungal cultures remain indispensable to determine the dermatophyte species…”

Response

“Depending on the domestic animal, a positive DME gives an idea of the fungal species and alerts to the risk of contracting dermatophytosis for humans in contact with the animal” in the Abstract was changed by

“A positive DME in an animal hair or skin sample should alert people in contact with the animal to the risk of contracting dermatophytosis.”

2

There is contradictory information about the study period: in Material and Methods. Animal samples it is referred to as 2010 to 2022, but in the Abstract, Results, and Tables 1,2  the authors cite 2008 and 2022.

2010 was corrected and changed by 2008 on page 2 (line 55): It was a mistake

3

The manuscript and results are very clear even a great variety of animal species were studied. However, I suggest indicating which animals were considered pets (for companionship or pleasure) and which were domestic animals to produce food or utility. The degree of contact between a pet owner and his animal is supposed to be higher than between a worker and animals for utility. So, the risk of acquiring a dermatophytid infection is more significant with a pet. It would be interesting a discussion about the risk of farm workers being infected by domesticated livestock (horses, cattle, and other animals for utility).

Response

We agree that the degree of contact between a pet owner and his or her animal is greater than between a worker and utility animals. However, if contact is made with an infected animal, the risk of contracting a dermatophyte infection is the same with a pet or a livestock animal. For instance, T. verrucosum infections are often identified in farmers. In fact, the risk could also depend on the dermatophyte species. Therefore, we did not discuss this item in the manuscript.

4

The unit system must be corrected since “ml” is wrong. According to
International System of Units, “L” is for liters, and “l” is for length. The entire
manuscript should be reviewed.

Response

Correction done as requested

5

Line 115: If 2745 samples were DME negative (Table 1) 64 represents 2.3% and not 10.5% as informed

6

Lines 116-118. Inform the percentages for the given number of isolates. These are the metrics that are needed.

Responses:

For both queries, the paragraph

Dermatophytes were isolated from 611 collected samples (Table 1). Positive DME was revealed in 547 (89.5%) of these samples. Only 64 (10.5%) of the dermatophyte cultures were obtained from samples when DME was negative. Further, 223 samples with positive DME did not generate a dermatophyte mycelium in cultures, and 2,681 samples were negative for both DME and dermatophyte growth.

was slightly modified as following:

Dermatophytes were isolated from 611 collected samples (Table 1). Positive DME was revealed in 547 (89.5%) of these samples. Only 64 dermatophytes (10.5%) were obtained from samples when DME was negative. Of a total of 3,515 samples analyzed, 223 samples with positive DME (6.3%) did not generate a dermatophyte mycelium in cultures, and 2,681 samples (76.3%) were negative for both DME and dermatophyte growth.

7

Line 127.  Does the data of routine analyses were published elsewhere? If it was not published or is not a Supplementary Material it may be informed as Data not shown.

Responses

The following senstence was inserted on line 133 :

PCR identification was performed on a total of 51 strains when identification based on phenotypic characters was doubtful or could be confused with a T. benhamiae of white phenotype.

8

Line 137. It would be of interest to inform the percentage of yellow-colored cultures found among T. benhamiae cultures.

Response:

Among the 50 dermatophytes isolated from guinea pigs, 48 were T. benhamiae, with a prevalence of the yellow phenotype (N = 43) in relation to the white phenotype (N = 5).

Inseted

Line 153. Delete a point before the reference. done

Line 175. Insert a point after T. done

Figure 1D needs to focus done: Fig 1D we changed

Figure 1. Delete “Samples were examined in a dissolving solution containing a fluorochrome that specifically binds fungal cell wall polysaccharides [11]” since it should be in Methods. Removed as requested

Table 3. In Column 1, delete the information “Dermatophytes species”, since it is not related to animals. Be sure that this information is in the Results section. Removed as requested

Reviewer 3 Report

Using a substantial number of hair and skin samples collected by veterinarians, the authors study the prevalence of dermatophytes in domestic animals in Switzerland. The study also indicates the usefulness of direct mycological examination by fluorescence microscopy as compared to fungal culture. This is a highly informative study, well presented and discussed. 

There is a missing semicolon on page 1, line 26 between "dermatophytosis" and "cats".

Author Response

Reviewer 3

Comments and Suggestions for Authors

Using a substantial number of hair and skin samples collected by veterinarians, the authors study the prevalence of dermatophytes in domestic animals in Switzerland. The study also indicates the usefulness of direct mycological examination by fluorescence microscopy as compared to fungal culture. This is a highly informative study, well presented and discussed. 

There is a missing semicolon on page 1, line 26 between "dermatophytosis" and "cats".

Response

“;” was added.

Reviewer 4 Report

I apologize in advance for the delay in completing the review.

I suggest changing the title. As the direct examination of clinical material for pathogenic fungi is routinely performed by veterinarians, the use of a fluorescent dye in routine diagnostics is rarely used. Even specialized laboratories rarely use such a technique.All the more so, the use of Blankophor as a dye is perhaps interesting but not really practical. This is due to the fact that it is practically not available in Europe. And from my own experience, an attempt to buy this substance is actually impossible - unless someone needs 1 ton of this substance right away. Hence my suggestion to change the title.

Another thing is that for many years, dermatophyte species have been "grouped" into complexes. Using the basic ITS sequence, we can only assign a given dermatophyte to a complex. It is a mistake to omit information to which complex a particular dermatophyte belongs. this is especially true of T. mentagrophytes.

Trichophyton mentagrophytes belongs to the T. mentagrophytes complex. As it cannot be distinguished from T. interdigitale without the use of several sequences, it is not wrong to write T. mentagrophytes/interdigitale complex. It cannot be assumed that because the dermatophyte was isolated from the animal, it means that it is T. mentagrophytes. What about reverse zoonosis?

Another element is the comparison of methods. The basic test is the use of a KOH and/or KOH + DMSO solution. Few people have a fluorescence microscope. I believe it should be used as a reference item. Only later is the comparison of the method with the fluorescent dye and the culture test. It is important to remember that finding elements of the fungus in the direct material does not mean that we will immediately obtain a pure culture from it.

When comparing the effectiveness of the method, assessed by ANOVA followed by HSD Tukey test should be used

Author Response

Reviewer 4

Comments and Suggestions for Authors

I apologize in advance for the delay in completing the review.

1

I suggest changing the title. As the direct examination of clinical material for pathogenic fungi is routinely performed by veterinarians, the use of a fluorescent dye in routine diagnostics is rarely used. Even specialized laboratories rarely use such a technique.All the more so, the use of Blankophor as a dye is perhaps interesting but not really practical. This is due to the fact that it is practically not available in Europe. And from my own experience, an attempt to buy this substance is actually impossible - unless someone needs 1 ton of this substance right away. Hence my suggestion to change the title.

Response:

The title was changed by

Survey on dermatophytes isolated from animals in Switzerland in the context of the prevention of zoonotic dermatophytosis.

2

Another thing is that for many years, dermatophyte species have been "grouped" into complexes. Using the basic ITS sequence, we can only assign a given dermatophyte to a complex. It is a mistake to omit information to which complex a particular dermatophyte belongs. this is especially true of T. mentagrophytes.

Trichophyton mentagrophytes belongs to the T. mentagrophytes complex. As it cannot be distinguished from T. interdigitale without the use of several sequences, it is not wrong to write T. mentagrophytes/interdigitale complex. It cannot be assumed that because the dermatophyte was isolated from the animal, it means that it is T. mentagrophytes. What about reverse zoonosis?

We used T. mentagrophytes for the species of the T. mentagrophytes/interdigitale complex. Many strains of this complex are indistinguishable on the basis of phenotypic characters, but Trichophyton interdigitale is almost always isolated from tinea pedis or tinea unguium and is a distinct species [17, 18].

This specification has been inserted on line 85

3

Another element is the comparison of methods. The basic test is the use of a KOH and/or KOH + DMSO solution. Few people have a fluorescence microscope. I believe it should be used as a reference item. Only later is the comparison of the method with the fluorescent dye and the culture test. It is important to remember that finding elements of the fungus in the direct material does not mean that we will immediately obtain a pure culture from it.

This was specified at the end of the Discussion section (lines X to Y)

When comparing the effectiveness of the method, assessed by ANOVA followed by HSD Tukey test should be used

Response

The objective of our survey was to review the dermatophytes isolated from animals in Switzerland in the context of the prevention of zoonotic dermatophytosis, and we used direct mycological examination and cultures for the identification of dermatophytes.

Interestingly, we found that the ratio of negative DME to positive DME was higher for M. canis [33/138 (23.9%)] than for T. mentagrophytes [24/328 (7.3%)]. We used a Chi-square test (the correct test in this situation) to show that these ratios were significantly different. Moreover, the title of the paper is now “Survey on dermatophytes isolated from animals in Switzerland in the context of the prevention of zoonotic dermatophytosis”

Round 2

Reviewer 4 Report

none